# On the Fundamental Trade-offs in Learning Invariant Representations

## Abstract

Many applications of representation learning, such as privacy-preservation, algorithmic fairness and domain adaptation, desire explicit control over semantic information being discarded. This goal is often formulated as satisfying two potentially competing objectives: maximizing utility for predicting a target attribute while simultaneously being independent or invariant with respect to a known semantic attribute. In this paper, we *identify and determine* two fundamental trade-offs between utility and semantic dependence induced by the statistical dependencies between the data and its corresponding target and semantic attributes. We derive closed-form solutions for the global optima of the underlying optimization problems under mild assumptions, which in turn yields closed formulae for the exact trade-offs. We also derive empirical estimates of the trade-offs and show their convergence to the corresponding population counterparts. Finally, we numerically quantify the trade-offs on representative problems and compare the solutions achieved by baseline representation learning algorithms.

## 1 Introduction

Real-world applications of representation learning algorithms often have to contend with objectives beyond predictive performance. These include cost functions pertaining to, invariance (e.g., to photometric or geometric variations), semantic independence (e.g., w.r.t to age or race for face recognition systems), privacy (e.g., mitigating leakage of sensitive information [1]), algorithmic fairness (e.g., demographic parity [2]), and generalization across multiple domains [3], to name a few.

At its core, the underlying goal of the aforementioned formulations of representation learning is to satisfy two competing objectives, extracting as much information necessary to predict a target label $y$ (e.g., face identity) while *intentionally* and *permanently* suppressing information pertaining to a desired semantic attribute $s$ (e.g., age, gender or race). When $y$ is independent of $s$, one can learn a representation that is independent of $s$ with no loss of performance, i.e., no trade-off exists between the two objectives. However, when the two attributes $y$ and $s$ are correlated, attaining semantic independence will necessarily reduce the performance of the target predictor, i.e., there is a trade-off between the two objectives. The trade-off is unknown yet is important for understanding the limits of existing and future representation learning algorithms that involve semantic independence constraints.

Let $z = f(x)$ be a representation of input data $x$, and $f(\cdot)$ be the encoder (see Fig 1(a)). Invariant learning requires that prediction of the target label, $\widehat{y} = g_Y(z)$ be independent of a semantic attribute $s$ i.e., $\widehat{y} \perp\!\!\!\perp s$ for all possible downstream target predictors $g_Y(\cdot)$. This independence condition is satisfied if and only if (iff), the representation $z$ is independent of $s$ i.e., $z \perp\!\!\!\perp s$. Therefore, Invariant representation learning (IRL) seeks to optimize two objectives: i) the degree of dependence between data representation $z$ and semantic attribute $s$, and ii) target task utility. These two objectives can be combined into one, with a parameter $\tau$ controlling the trade-off.

Submitted to 35th Conference on Neural Information Processing Systems (NeurIPS 2021). Do not distribute.

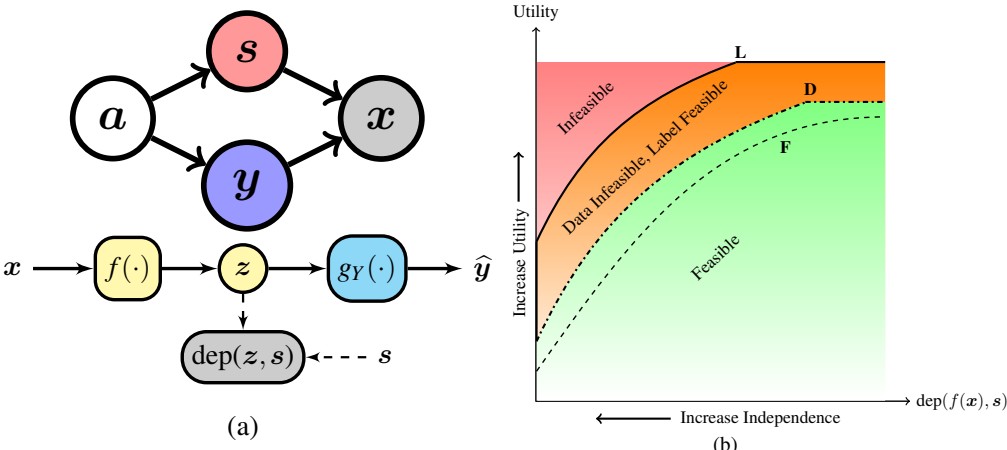

(a)                                          (b)

Figure 1: (a): Generic frame work of invariant representation learning (IRL) where attributes $s$ and $y$ are caused by a latent factor $a$ and are not marginally independent. Under this setting, IRL seeks a representation $z = f(x)$ that contains enough information for downstream target predictor $g_Y(\cdot)$ while being independent of the semantic attribute $s$. Consequently, the prediction $\hat{y} = g_Y(z)$ will also be independent of $s$ for any downstream predictor $g_Y(\cdot)$. (b): We identify and determine two different fundamental trade-offs between utility (i.e., the performance of target task predictor) and dependence measure $\text{dep}(z, s)$ by an optimal learner in the hypothesis class of Borel-measurable functions. Trade-off **L** is induced by the joint distribution of the labels $p_{ys}$. Trade-off **D** is induced by the joint distribution of the data $p_{xys}$. Trade-off **F** is a relaxed version of trade-off **D** obtained by either using a surrogate measure of dependence, e.g., adversarial learning [3] or from a constrained hypothesis class [4], or from using sub-optimal optimization algorithms.

37 In this paper, we identify and analytically determine two fundamental trade-offs in the invariant
38 representation learning setting introduced above, namely *Data Space Trade-Off* and *Label Space*
39 *Trade-Off*. These trade-offs are illustrated in Figure 1 (b) and formally defined next.

40 **Definition 1.** *Data Space Trade-Off* arises from the statistical dependence between the target attribute
41 $y$ and the semantic attribute $s$ conditioned on the given input data $x$. When the learner's hypothesis
42 class contains all Borel-measurable functions[1] we have:

$$\inf_{f(\cdot) \text{ measurable}} \left\{ (1-\tau) \inf_{g_Y(\cdot) \text{ measurable}} \mathbb{E}_{x,y}\left[ \mathcal{L}_Y\Big(g_Y(f(x)), y\Big) \right] + \tau \, \text{dep}(f(x), s) \right\}. \quad (1)$$

43 where $f(\cdot)$ is the encoder that extracts representation $z$ from $x$, $g_Y(\cdot)$ predicts $\hat{y}$ from the repre-
44 sentation $z$, $\mathcal{L}_Y(\cdot, \cdot)$ is the loss for the desired task of predicting the task label $y$. The function
45 $\text{dep}(\cdot, \cdot) \geq 0$ is a parametric or non-parametric measure of statistical dependence i.e., $\text{dep}(q, r) = 0$
46 means $q$ and $r$ are independent, and $\text{dep}(q, r) > 0$ means $q$ and $r$ are dependent with larger values
47 indicating greater degrees of dependence. The scalar $\tau \in [0, 1)$ is a hyper-parameter that controls
48 the trade-off between the two objectives, with $\tau = 0$ being the standard approach that enforces no
49 independence to the attribute $s$, while $\tau \to 1$ enforces representation $z$ to be independent of $s$.

50 Including all measurable functions in the hypothesis class of the encoder $f(\cdot)$ and target predic-
51 tor $g_Y(\cdot)$ ensures that the best possible trade-off is included within the feasible solution space.
52 For example, when $\tau = 0$ and $\mathcal{L}_Y(\cdot, \cdot)$ is the mean-squared error, the optimal Bayes estimator,
53 $g_Y(f(x)) = \mathbb{E}_y[y \,|\, x]$ is reachable. This definition corresponds to the trade-off **D** in Figure 1 (b).

54 **Definition 2.** *Label Space Trade-Off* arises by ignoring the data $x$ and is purely determined by the
55 statistical dependence between the target feature $y$ and the semantic attribute $s$. Such a trade-off can
56 be defined as:

$$\inf_{z \in L^2} \left\{ (1-\tau) \inf_{g_Y(\cdot) \text{ measurable}} \mathbb{E}_{x,y}\left[ \mathcal{L}_Y(g_Y(z), y) \right] + \tau \, \text{dep}(z, s) \right\}, \quad (2)$$

57 where $L^2$ is the space of all random vectors with finite second-order moment (i.e., $\mathbb{E}_z[\|z\|^2] < \infty$)
58 on the same probability space in which the joint variable $(s, y)$ comes from.

---

[1]More specifically, we consider square-integrable Borel-measurable functions for boundedness.

This definition corresponds to the optimal trade-off obtained by an *ideal* representation $z$ that is not constrained by the learnability of the encoder $f(\cdot)$. For example, if $\tau = 0$, the ideal representation $z$ is perfectly aligned with the target label $y$ i.e., $z = y$ and $g_Y(\cdot)$ is the identity function, perfect prediction of target attribute is feasible. Therefore, this trade-off corresponds to the best trade-off that any combination of data $x$ and learnable encoder $f(\cdot)$ can aspire to. This definition corresponds to the trade-off **L** in Figure 1 (b), and it necessarily dominates the *Data Space Trade-Off* **D**.

**Contributions:** i) Identify two fundamental trade-offs in invariant representation learning. ii) Obtain closed-form solution for the corresponding optimization problems, and consequently determine the trade-offs exactly. iii) Provide consistent empirical closed-form solution for the representations that achieve optimal trade-offs. iv) Numerically quantify the trade-offs defined here and compare them to those obtained by existing solutions.

**Implications:** i) Our closed-form empirical estimators for the optimal representations lend themselves to practical invariant representation learning algorithms. ii) Theoretically elucidating and empirically quantifying the intrinsic limits of invariant representations will enable researchers and practitioners alike to identify the feasible and infeasible solution space for the trade-offs and lead to informed development and deployment of optimal IRL methods. iii) Our theoretical analysis sheds light on the utility-semantic independence trade-off, the role of statistical dependency between target label $y$, the semantic attribute $s$, and the input data $x$, and the hypothesis class adopted for the learners.

## 2 Related Work

**Trade-Offs in Representation Learning:** While there are abundant empirical approaches for the representation learning applications considered in this paper, to the best of our knowledge, there is no prior work that *exactly* characterizes and empirically quantifies the trade-offs inherent to representation learning with semantic independence constraints.

Prior work primarily sought to either obtain lower or upper bounds or characterize the extreme points of the trade-off in specific contexts such as fair representation learning. For instance, [5] uses information theoretic tools and characterizes the utility-fairness trade-off in terms of a lower bounds when both $y$ and $s$ are binary labels. Later [6] provided both upper and lower bound for the binary labels. By leveraging Chernoff bound [7] proposed a construction method to generate an ideal representation beyond input data to achieve perfect fairness while maintaining the best performance on target task for equalized odds. In the case of categorical features, a lower bound on utility-fairness trade-off has been provided by [8]. The notion of Pareto optimality was used by [9] to minimize the maximum possible error among sensitive attributes where both target and sensitive features are categorical. In contrast to this body of work, our trade-off analysis is applicable to multi-dimensional discrete and/or continuous attributes where we find the exact optimal trade-offs.

The only prior work that investigates fundamental trade-offs in a general setting where both $y$ and $s$ can be continuous or discrete features, are [4] and [10]. [4] considers only linear dependence between the representation and semantic attribute and proposed a closed-form solution for the utility-fairness trade-off. Even though [10] considers non-linear dependencies, optimal losses have been derived only for the extremes of the trade-off (i.e., $\tau \to 0$ and $\tau \to 1$). In a more general setting where $0 < \tau < 1$, [10] only provides a lower bound on utility-invariance trade-off through information plane analysis. In contrast to the foregoing, we take a functional analysis approach and utilize covariance operator based measures of dependence that account for all non-linear dependence relations. We exactly characterize and quantify the utility-invariance trade-offs, while also providing a means to empirically estimate the encoder that achieves said optimal trade-off. Lastly, in addition to the *Data Space Trade-Off*, we also introduce and determine the *Label Space Trade-Off* which is the ideal trade-off that any unrestricted learning algorithm can aspire to.

**Invariant, Fair, Privacy-Preserving Representation Learning**: The basic idea of representation learning that discards unwanted semantic information has been explored under different contexts like invariant, fair, or privacy-preserving learning. In domain adaptation [11, 12, 13], the goal is to learn features that are independent of the data domain. In fair learning [14, 15, 16, 17, 18, 19, 20, 21, 22, 23, 2, 24, 25, 26, 27, 4], the goal is to discard the demographic information that leads to unfair outcomes. Similarly, there is a growing interest in mitigating unintended leakage of private information from data representations [28, 29, 1, 30, 31]. A vast majority of this body of work is empirical in nature. These methods implicitly look for a single or more points in the trade-off between utility and fairness

and do not explicitly seek to characterize the whole trade-off front. Overall, these approaches are not concerned (or aware) about the feasibility and limitations on the utility-invariance trade-off. In contrast, this paper determines the fundamental theoretical limits of controlling independence to semantic attributes, and proposes practical learning algorithms that achieve this limit.

**Adversarial Representation Learning:** Most practical approaches for learning fair, invariant, domain adaptive or privacy-preserving representations discussed above are based on adversarial representation learning (ARL). This learning problem is typically formulated as,

$$\inf_{f \in \mathcal{H}_x} \left\{ (1 - \tau) \inf_{g_Y \in \mathcal{H}_y} \mathbb{E}_{\boldsymbol{x}, \boldsymbol{y}} \Big[ \mathcal{L}_Y \Big( g_Y(f(\boldsymbol{x})), \boldsymbol{y} \Big) \Big] - \tau \inf_{g_S \in \mathcal{H}_s} \mathbb{E}_{\boldsymbol{x}, \boldsymbol{s}} \Big[ \mathcal{L}_S \Big( g_S(f(\boldsymbol{x})), \boldsymbol{s} \Big) \Big] \right\}, \quad (3)$$

where $\mathcal{L}_S(\cdot, \cdot)$ is the loss function of a hypothetical adversary $g_S(\cdot)$ who intends to extract the semantic attribute $\boldsymbol{s}$ through the best predictor within the hypothesis class $\mathcal{H}_s$. ARL is a special case of the *Data Space Trade-Off* in (1) where the negative loss of the adversary, $- \inf_{g_S \in \mathcal{H}_s} \mathbb{E}_{\boldsymbol{x}, \boldsymbol{s}} \Big[ \mathcal{L}_S \Big( g_S(f(\boldsymbol{x})), \boldsymbol{s} \Big) \Big]$ plays the role of $\text{dep}(\boldsymbol{f}(\boldsymbol{x}), \boldsymbol{s})$. However, this form of adversarial learning suffers from a fundamental drawback as also noted in [32, 33]. The measure of dependence induced by ARL does not account for all modes of non-linear dependence between $\boldsymbol{s}$ and the representation $\boldsymbol{z}$. The next theorem states this observation precisely,

**Theorem 1.** [2] Let $\mathcal{H}_{\boldsymbol{s}}$ contain all Borel-measurable functions and $\mathcal{L}_S(\cdot, \cdot)$ be mean squared error (MSE) loss. Then,

$$\boldsymbol{z} \in \arg\sup \left\{ \inf_{g_S \in \mathcal{H}_s} \mathbb{E}_{\boldsymbol{x}, \boldsymbol{s}} \Big[ \mathcal{L}_S \Big( g_S(\boldsymbol{z}), \boldsymbol{s} \Big) \Big] \right\} \Leftrightarrow \mathbb{E}[\boldsymbol{s} \,|\, \boldsymbol{z}] = \mathbb{E}[\boldsymbol{s}].$$

This theorem implies that an optimal adversary does not necessarily lead to a representation $\boldsymbol{z}$ that is statistically independent of $\boldsymbol{s}$ (i.e., $p(\boldsymbol{s}|\boldsymbol{z}) = p(\boldsymbol{s})$), but rather leads to $\boldsymbol{s}$ being mean independent of representation $\boldsymbol{z}$ i.e., independence with respect to first order moment only. In other words, adversarially learned measure of dependence is not a complete measure of dependence and hence does not account for all modes of non-linear dependence between two random variables. As such, ARL is inherently incapable of attaining the trade-offs achievable by complete measures of dependence.

## 3 Theoretical Results

### 3.1 Problem Setting

Consider the probability space $(\Omega, \mathcal{F}, \mathbb{P})$, where $\Omega$ is the sample space, $\mathcal{F}$ is a $\sigma-$algebra on $\Omega$, and $\mathbb{P}$ is a probability measure on $\mathcal{F}$. We assume that the joint random vector $(\boldsymbol{x}, \boldsymbol{y}, \boldsymbol{s})$, containing the input data $\boldsymbol{x} \in \mathbb{R}^{d_x}$, the target label $\boldsymbol{y} \in \mathbb{R}^{d_y}$ and the sensitive attribute $\boldsymbol{s} \in \mathbb{R}^{d_s}$, is a random vector on $(\Omega, \mathcal{F})$ with joint distribution $p_{\boldsymbol{xys}}$.

**Assumption 1.** We assume that the encoder consists of $r$ functions in an $L_2$-universal RKHS $(\mathcal{H}_{\boldsymbol{x}}, k_{\boldsymbol{x}}(\cdot, \cdot))$ (e.g., Gaussian kernel), where $L_2-$universality guarantees that $\mathcal{H}_{\boldsymbol{x}}$ can approximate any Borel-measurable function with arbitrary precision [34].

Now, the representation vector $\boldsymbol{z}$ can be expressed as

$$\boldsymbol{z} = \boldsymbol{f}(\boldsymbol{x}) := \Big[ f_1(\boldsymbol{x}), \cdots, f_r(\boldsymbol{x}) \Big]^T \in \mathbb{R}^r, \quad f_j(\cdot) \in \mathcal{H}_{\boldsymbol{x}} \,\forall j = 1, \ldots, r. \quad (4)$$

where $r$ is the dimensionality of the representation $\boldsymbol{z}$. As discussed in Corollary 5.1, unlike common practice where it is chosen arbitrarily, $r$ itself is an object of interest for optimization. We consider a general scenario where both $\boldsymbol{y}$ and $\boldsymbol{s}$ can be continuous or discrete, or one of $\boldsymbol{y}$ or $\boldsymbol{s}$ is continuous while the other is discrete. To do this, we substitute[3] the target loss, $\inf_{g_Y} \mathbb{E}_{\boldsymbol{x}, \boldsymbol{y}}[\mathcal{L}_Y(g_Y(\boldsymbol{z}), \boldsymbol{y})]$ in (1) with the negative of a non-parametric measure of dependence i.e., $-\text{dep}(\boldsymbol{z}, \boldsymbol{y})$. Furthermore, in

---

[2]We defer the proofs of all lemmas, theorems and corollaries to the supplementary material.

[3]Many standard loss functions can be written in term of dependence measures [35] that capture all non-linear dependencies i.e, $\mathbb{E}_{\boldsymbol{x}, \boldsymbol{y}} \big[ \mathcal{L}_Y \big( f_T(\boldsymbol{f}(\boldsymbol{x})), \boldsymbol{y} \big) \big] \propto -\text{dep}(\boldsymbol{f}(\boldsymbol{x}), \boldsymbol{y})$. For example, the mean squared error is proportional to $1 - \rho(f(\boldsymbol{x}), \boldsymbol{y})$, where $\rho$ is the Pearson correlation coefficient, a plausible dependence measure.

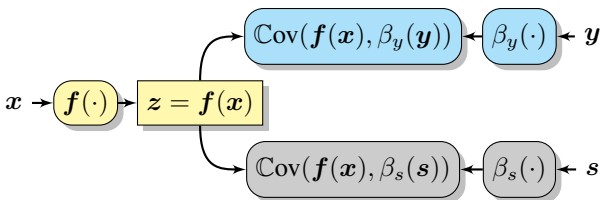

Figure 2: Our IRL model consists of three components: i) An $r$-dimensional encoder $\boldsymbol{f}(\cdot)$ in a RKHS $\mathcal{H}_{\boldsymbol{x}}$. ii) A measure of dependence that accounts for all kinds of linear or non-linear dependencies between the representation $\boldsymbol{z}$ and the semantic attribute $\boldsymbol{s}$ via the covariance between $\boldsymbol{f}(\boldsymbol{x})$ and $\beta_s(\boldsymbol{s})$ where $\boldsymbol{x}$ is the input data and $\beta_s(\cdot)$ belongs to RKHS $\mathcal{H}_{\boldsymbol{s}}$. iii) A measure of dependency between $\boldsymbol{f}(\boldsymbol{x})$ and the target attribute $\boldsymbol{y}$ defined similar to the one for $\boldsymbol{s}$.

unsupervised settings, when there is no target attribute $\boldsymbol{y}$, the target dependence $\mathrm{dep}(\boldsymbol{z}, \boldsymbol{y})$ can be replaced with $\mathrm{dep}(\boldsymbol{z}, \boldsymbol{x})$, which implicitly forces the representation $\boldsymbol{z}$ to retain as much information as is necessary for reconstructing the input data $\boldsymbol{x}$. This scenario is of practical interest when a data producer aims to provide a representation of data that is independent of a desired semantic attribute for any arbitrary downstream task.

We start by designing $\mathrm{dep}(\boldsymbol{z}, \boldsymbol{s})$, and $\mathrm{dep}(\boldsymbol{z}, \boldsymbol{y})$ follows similarly. A key desiderata of dependence measures is that they should be able to account for all possible non-linear dependence relations between the random variables (or vectors). Examples of such measures include information theoretic measures such as mutual information (e.g., MINE [36]) or covariance operator based measures such as Hilbert-Schmidt Independence Criterion [37], Constrained Covariance [38] and Kernel Canonical Correlation [39]. The underlying principle behind the latter class of dependence measures is that finite dimensional spaces with non-linear dependencies behave as linearly dependent spaces when mapped appropriately to higher dimensional spaces. In this paper we adopt the covariance operator based measures as our choice of dependence measure for analytical tractability.

Principally, $\boldsymbol{z}$ and $\boldsymbol{s}$ are independent iff $\mathbb{Cov}(\alpha(\boldsymbol{z}), \beta_s(\boldsymbol{s}))$ is zero for all $\alpha(\cdot)$ and $\beta_s(\cdot)$ belonging to some universal RKHSs [38]. Since $\boldsymbol{z} = \boldsymbol{f}(\boldsymbol{x})$ and $f(\cdot) \in \mathcal{H}_x$, $\mathbb{Cov}(\alpha(\boldsymbol{z}), \beta_s(\boldsymbol{s})) = \mathbb{Cov}(\alpha(\boldsymbol{f}(\boldsymbol{x})), \beta_s(\boldsymbol{s}))$, which necessitates application of a kernel on top of another kernel. This limits the analytical tractability of our solution. However, as we argue below, it is almost sufficient to consider transformation on $\boldsymbol{s}$, only, in which case it reduces to $\mathbb{Cov}(\boldsymbol{f}(\boldsymbol{x}), \beta_s(\boldsymbol{s}))$. Let $(\mathcal{H}_{\boldsymbol{s}}, k_{\boldsymbol{s}}(\cdot, \cdot))$ and $(\mathcal{H}_{\boldsymbol{y}}, k_{\boldsymbol{y}}(\cdot, \cdot))$ be separable[4] RKHSs of functions defined on $\mathbb{R}^{d_s}$ and $\mathbb{R}^{d_y}$, respectively. Consider the bi-linear functional,

$$h(\cdot, \cdot) : \mathcal{H}_{\boldsymbol{x}} \times \mathcal{H}_{\boldsymbol{s}} \to \mathbb{R}, \quad h_j(f_j, \beta_s) := \mathbb{Cov}_{\boldsymbol{x}, \boldsymbol{s}}(f_j(\boldsymbol{x}), \beta_s(\boldsymbol{s})). \tag{5}$$

**Assumption 2.** We assume in the rest of this paper that the positive definite kernel functions are bounded, i.e.,

$$\mathbb{E}_{\boldsymbol{x}}[k_{\boldsymbol{x}}(\boldsymbol{x}, \boldsymbol{x})] < \infty, \quad \mathbb{E}_{\boldsymbol{s}}[k_{\boldsymbol{s}}(\boldsymbol{s}, \boldsymbol{s})] < \infty, \quad \text{and} \quad \mathbb{E}_{\boldsymbol{y}}[k_{\boldsymbol{y}}(\boldsymbol{y}, \boldsymbol{y})] < \infty. \tag{6}$$

The assumptions in (6) guarantee that $h(\cdot, \cdot)$ in (5) is bounded [40] and therefore, invoking Riesz representation theorem [41], there exists a unique and bounded linear operator $\Sigma_{\boldsymbol{s}\boldsymbol{x}}$, such that

$$h(f, \beta_s) = \mathbb{Cov}_{\boldsymbol{x}, \boldsymbol{s}}(f(\boldsymbol{x}), \beta_s(\boldsymbol{s})) = \langle \beta_s, \Sigma_{\boldsymbol{s}\boldsymbol{x}} f \rangle_{\mathcal{H}_{\boldsymbol{s}}} \quad \forall f \in \mathcal{H}_{\boldsymbol{x}}, \forall \beta_s \in \mathcal{H}_{\boldsymbol{s}}. \tag{7}$$

Based on $h(\cdot, \cdot)$, we define the linear operator $\boldsymbol{h}_{\boldsymbol{f}, \boldsymbol{s}} : \mathcal{H}_{\boldsymbol{s}} \to \mathbb{R}^r$ as

$$\boldsymbol{h}_{\boldsymbol{f}, \boldsymbol{s}}(\beta_s) := \begin{bmatrix} \mathbb{Cov}_{\boldsymbol{x}, \boldsymbol{s}}(f_1(\boldsymbol{x}), \beta_s(\boldsymbol{s})) \\ \vdots \\ \mathbb{Cov}_{\boldsymbol{x}, \boldsymbol{s}}(f_r(\boldsymbol{x}), \beta_s(\boldsymbol{s})) \end{bmatrix} = \begin{bmatrix} \langle \beta_s, \Sigma_{\boldsymbol{s}\boldsymbol{x}} f_1 \rangle_{\mathcal{H}_{\boldsymbol{s}}} \\ \vdots \\ \langle \beta_s, \Sigma_{\boldsymbol{s}\boldsymbol{x}} f_r \rangle_{\mathcal{H}_{\boldsymbol{s}}} \end{bmatrix}.$$

The operator $\boldsymbol{h}_{\boldsymbol{f}, \boldsymbol{s}}$ captures all modes of non-linear dependence, since the distribution of a low-dimensional projection of high-dimensional data is approximately normal [42], [43]. In other words, we assume that $(\boldsymbol{f}(\boldsymbol{x}), \beta_s(\boldsymbol{s}))$ is an approximately Gaussian random vector.

---

[4]By separable we mean having a countable orthonormal basis set.

Among the different dependence measures that have been defined through the covariance operator we adopt the Hilbert-Schmidt Independence Criterion (HSIC) [37] which is defined as the Hilbert-Schmidt norm (HS-norm) of the covariance operator,

$$\text{dep}(\boldsymbol{z}, \boldsymbol{s}) := \|\boldsymbol{h}_{\boldsymbol{f}, \boldsymbol{s}}\|_{\text{HS}}^2 = \sum_{\beta_s \in \mathcal{U}_{\boldsymbol{s}}} \|\boldsymbol{h}_{\boldsymbol{f}, \boldsymbol{s}}(\beta_s)\|_2^2 = \sum_{\beta_s \in \mathcal{U}_{\boldsymbol{s}}} \sum_{j=1}^{r} h^2(f_j, \beta_s) \tag{8}$$

where $\mathcal{U}_{\boldsymbol{s}}$ is a countable orthonormal basis set for $\mathcal{H}_{\boldsymbol{s}}$. Note that, based on this definition, if the distribution $(\boldsymbol{f}(\boldsymbol{x}), \beta_s(\boldsymbol{s}))$ fails to be a normal distribution, we end up measuring mean dependency of $\boldsymbol{z} = \boldsymbol{f}(\boldsymbol{x})$ from $\boldsymbol{s}$ which is still much stronger than the linear dependency between $\boldsymbol{z}$ and $\boldsymbol{s}$ [44]. Even under this assumption, empirically (Section 4) we observe that trade-offs we obtain significantly dominate those from existing invariant representation learning algorithms.

The following Lemma introduces a well-defined population expression for $\text{dep}(\boldsymbol{z}, \boldsymbol{s})$ in (8).

**Lemma 2.**

$$\begin{aligned}
\text{dep}(\boldsymbol{z}, \boldsymbol{s}) \quad = \quad \sum_{j=1}^{r} \Big\{ &\mathbb{E}_{\boldsymbol{x}, \boldsymbol{s}, \boldsymbol{x}', \boldsymbol{s}'} \Big[ f_j(\boldsymbol{x}) \, f_j(\boldsymbol{x}') \, k_{\boldsymbol{s}}(\boldsymbol{s}, \boldsymbol{s}') \Big] + \mathbb{E}_{\boldsymbol{x}}[f_j(\boldsymbol{x})] \mathbb{E}_{\boldsymbol{x}'}[f_j(\boldsymbol{x}')] \, \mathbb{E}_{\boldsymbol{s}, \boldsymbol{s}'}[k_{\boldsymbol{s}}(\boldsymbol{s}, \boldsymbol{s}')] \\
&- 2 \, \mathbb{E}_{\boldsymbol{x}, \boldsymbol{s}} \Big[ f_j(\boldsymbol{x}) \, \mathbb{E}_{\boldsymbol{x}'}[f_j(\boldsymbol{x}')] \, \mathbb{E}_{\boldsymbol{y}'}[k_{\boldsymbol{s}}(\boldsymbol{s}, \boldsymbol{s}')] \Big] \Big] \Big\}
\end{aligned}$$

where $(\boldsymbol{x}, \boldsymbol{s})$ and $(\boldsymbol{x}', \boldsymbol{s}')$ are independently drawn from the joint distribution $\boldsymbol{p}_{\boldsymbol{xs}}$.

In practice, it is necessary to empirically estimate $\text{dep}(\boldsymbol{z}, \boldsymbol{s})$, since the population distributions are typically unknown in most real-world scenarios.

**Definition 3.** Let $\boldsymbol{D} = \{(\boldsymbol{x}_1, \boldsymbol{s}_1, \boldsymbol{y}_1), \cdots, (\boldsymbol{x}_n, \boldsymbol{s}_n, \boldsymbol{y}_n)\}$ be the training data, containing $n$ i.i.d. realizations from the joint distribution $\boldsymbol{p}_{\boldsymbol{xsy}}$. Using, the representer theorem [45], it follows that $\boldsymbol{f}(\boldsymbol{x}) = \boldsymbol{\Theta}_E[k_{\boldsymbol{x}}(\boldsymbol{x}_1, \boldsymbol{x}), \cdots, k_{\boldsymbol{x}}(\boldsymbol{x}_n, \boldsymbol{x})]^T$, where $\boldsymbol{\Theta} \in \mathbb{R}^{r \times n}$ is a free parameter matrix.

**Lemma 3.** Let an empirical estimation of covariance be

$$\mathbb{C}\text{ov}_{\boldsymbol{x}, \boldsymbol{s}}(f_j(\boldsymbol{x}), \beta_s(\boldsymbol{s})) \approx \frac{1}{n} \sum_{i=1}^{n} f_j(\boldsymbol{x}_i) \beta_s(\boldsymbol{s}_i) - \frac{1}{n^2} \sum_{i=1}^{n} \sum_{k=1}^{n} f_j(\boldsymbol{x}_i) \beta_s(\boldsymbol{s}_k).$$

Then, the empirical estimator of $\text{dep}(\boldsymbol{z}, \boldsymbol{s})$ is given by

$$\text{dep}^{\text{emp}}(\boldsymbol{z}, \boldsymbol{s}) \quad := \quad \frac{1}{n^2} \|\boldsymbol{\Theta} \boldsymbol{K}_{\boldsymbol{x}} \boldsymbol{H} \boldsymbol{L}_{\boldsymbol{s}}\|_F^2, \tag{9}$$

where $\boldsymbol{K}_{\boldsymbol{x}}, \boldsymbol{K}_{\boldsymbol{s}} \in \mathbb{R}^{n \times n}$ are Gram matrices corresponding to $\mathcal{H}_{\boldsymbol{x}}$ and $\mathcal{H}_{\boldsymbol{s}}$, respectively, $\boldsymbol{H} = \boldsymbol{I} - \frac{1}{n} \boldsymbol{1} \boldsymbol{1}^T$, and $\boldsymbol{L}_{\boldsymbol{s}}$ is a full column-rank matrix in which $\boldsymbol{L}_{\boldsymbol{s}} \boldsymbol{L}_{\boldsymbol{s}}^T = \boldsymbol{K}_{\boldsymbol{s}}$ (Cholesky factorization). This empirical estimator in (9) has a bias of $\mathcal{O}(n^{-1})$ and a convergence rate of $\mathcal{O}(n^{-1/2})$.

The population and empirical dependence measures between $\boldsymbol{z}$ and $\boldsymbol{y}$ i.e., $\text{dep}(\boldsymbol{z}, \boldsymbol{y})$ and $\text{dep}^{\text{emp}}(\boldsymbol{z}, \boldsymbol{y})$, respectively, can be defined and obtained similarly.

## 3.2 Trade-Off D

We now turn to the the optimization problem corresponding to the trade-off **D** in (1). Recall that $\boldsymbol{z} = \boldsymbol{f}(\boldsymbol{x})$ is $r$-dimensional, where the dimensionality $r$ is a free variable. A common desiderata of learned representations is that of compactness [46] in order to avoid learning representations with redundant information where different dimensions are highly correlated with each other. Therefore, going beyond the assumption that each component of $\boldsymbol{f}(\cdot)$ (i.e., $f_j(\cdot)$) belongs to a $L_2-$universal RKHS $\mathcal{H}_{\boldsymbol{x}}$, we impose additional constraints on the representation. Specifically, we constrain the search space of the encoder $\boldsymbol{f}(\cdot)$ to learn a disentangled representation [46] as follows,

$$\mathcal{A}_r := \Big\{ \big(f_1(\cdot), \cdots, f_r(\cdot)\big) \, \Big| \, f_i, f_j \in \mathcal{H}_{\boldsymbol{x}}, \, \mathbb{C}\text{ov}_{\boldsymbol{x}}(f_i(\boldsymbol{x}), f_j(\boldsymbol{x})) + \gamma \langle f_i, f_j \rangle_{\mathcal{H}_{\boldsymbol{x}}} = \delta_{i,j} \Big\}, \tag{10}$$

where the regularization term $\gamma \langle f_i, f_j \rangle_{\mathcal{H}_{\boldsymbol{x}}}$, encourages orthogonality and boundedness, which in turn forces the representation to be compact or non-redundant. Such disentangled representations have

been studied in the context of independent component analysis (ICA) [39]. Now, the optimization problem in (1) reduces to,

$$\sup_{\boldsymbol{f} \in \mathcal{A}_r} \Big\{ J(\boldsymbol{f}(\boldsymbol{x})) := (1 - \tau) \operatorname{dep}(\boldsymbol{f}(\boldsymbol{x}), \boldsymbol{y}) - \tau \operatorname{dep}(\boldsymbol{f}(\boldsymbol{x}), \boldsymbol{s}) \Big\}, \quad 0 \leq \tau < 1, \tag{11}$$

where as justified earlier the target loss function $\inf_{f_Y} \mathbb{E}_{\boldsymbol{x}, \boldsymbol{y}}[\mathcal{L}_Y(f_T(\boldsymbol{f}(\boldsymbol{x})), \boldsymbol{y})]$ is substituted by $-\operatorname{dep}(\boldsymbol{f}(\boldsymbol{x}), \boldsymbol{y})$. Fortunately, the above optimization problem lends itself to a closed-form solution as given by the next theorem.

**Theorem 4.** A solution[5] to the optimization problem in (11) is the eigenfunctions corresponding to $r$ largest eigenvalues of the following generalized problem

$$\Big( (1 - \tau) \Sigma_{\boldsymbol{y}\boldsymbol{x}}^* \Sigma_{\boldsymbol{y}\boldsymbol{x}} - \tau \Sigma_{\boldsymbol{s}\boldsymbol{x}}^* \Sigma_{\boldsymbol{s}\boldsymbol{x}} \Big) f = \lambda \Sigma_{\boldsymbol{x}\boldsymbol{x}} f, \tag{12}$$

where $\Sigma_{\boldsymbol{s}\boldsymbol{x}}$ and $\Sigma_{\boldsymbol{y}\boldsymbol{x}}$ are the covariance operators defined in (7), and $\Sigma_{\boldsymbol{s}\boldsymbol{x}}^*$ and $\Sigma_{\boldsymbol{y}\boldsymbol{x}}^*$ are the adjoint operators of $\Sigma_{\boldsymbol{s}\boldsymbol{x}}$ and $\Sigma_{\boldsymbol{y}\boldsymbol{x}}$, respectively.

**Remark.** If the trade-off parameter $\tau = 0$ (i.e., no semantic independence constraint is imposed), the solution in Theorem 4 resembles a supervised version of ICA in [39] which is essentially a kernelized dimensionality reduction supervised by the target attribute $\boldsymbol{y}$. On the other hand, if $\tau \to 1$ (i.e., utility is ignored and only semantic independence is considered), the solution in Theorem 4 is the eigenfunctions corresponding to the negative eigenvalues of $\Sigma_{\boldsymbol{s}\boldsymbol{x}}^* \Sigma_{\boldsymbol{s}\boldsymbol{x}}$, which are the directions that are least explanatory of the semantic attribute $\boldsymbol{s}$.

An empirical version of (11) is the following optimization problem

$$\sup_{\boldsymbol{f} \in \mathcal{A}_r} \Big\{ J^{\mathrm{emp}}(\boldsymbol{f}(\boldsymbol{x})) := (1 - \tau) \operatorname{dep}^{\mathrm{emp}}(\boldsymbol{f}(\boldsymbol{x}), \boldsymbol{y}) - \tau \operatorname{dep}^{\mathrm{emp}}(\boldsymbol{f}(\boldsymbol{x}), \boldsymbol{s}) \Big\}, \quad 0 \leq \tau < 1 \tag{13}$$

where $\operatorname{dep}^{\mathrm{emp}}(\boldsymbol{f}(\boldsymbol{x}), \boldsymbol{s})$ and $\operatorname{dep}^{\mathrm{emp}}(\boldsymbol{f}(\boldsymbol{x}), \boldsymbol{y})$ are given in (9).

**Theorem 5.** Consider the Cholesky factorization $\boldsymbol{K}_{\boldsymbol{x}} = \boldsymbol{L}_{\boldsymbol{x}} \boldsymbol{L}_{\boldsymbol{x}}^T$, where $\boldsymbol{L}_{\boldsymbol{x}}$ is a full column-rank matrix. A solution to (13) is

$$\boldsymbol{f}^{\mathrm{opt}} = \boldsymbol{\Theta}^{\mathrm{opt}} \Big[ k_{\boldsymbol{x}}(\boldsymbol{x}_1, \cdot), \cdots, k_{\boldsymbol{x}}(\boldsymbol{x}_n, \cdot) \Big]^T$$

where $\boldsymbol{\Theta}^{\mathrm{opt}} = \boldsymbol{U}^T (\boldsymbol{L}_{\boldsymbol{x}})^\dagger$ and the columns of $\boldsymbol{U}$ are eigenvectors corresponding to $r$ largest eigenvalues, $\lambda_1, \cdots, \lambda_r$ of the following generalized problem,

$$\Big( \boldsymbol{L}_{\boldsymbol{x}}^T ((1 - \tau) \tilde{\boldsymbol{K}}_y - \tau \tilde{\boldsymbol{K}}_s) \boldsymbol{L}_{\boldsymbol{x}} \Big) \boldsymbol{u} = \lambda \Big( \boldsymbol{L}_{\boldsymbol{x}}^T \boldsymbol{H} \boldsymbol{L}_{\boldsymbol{x}} + n\gamma \boldsymbol{I} \Big) \boldsymbol{u} \tag{14}$$

where $\gamma$ is the regularization parameter from (10) and the supremum value of (13) is $\sum_{j=1}^r \lambda_j$.

**Corollary 5.1.** *Embedding Dimensionality*: A useful corollary of Theorem 5 is optimal embedding dimensionality:

$$\arg\sup_r \Bigg\{ \sup_{\boldsymbol{f} \in \mathcal{A}_r} \Big\{ J^{\mathrm{emp}}(\boldsymbol{f}(\boldsymbol{x})) := (1 - \tau) \operatorname{dep}^{\mathrm{emp}}(\boldsymbol{f}(\boldsymbol{x}), \boldsymbol{y}) - \tau \operatorname{dep}^{\mathrm{emp}}(\boldsymbol{f}(\boldsymbol{x}), \boldsymbol{s}) \Big\} \Bigg\},$$

which is the number of positive eigenvalues of the generalized eigenvalue problem in (14). To intuitively examine this result, consider two extreme cases: i) If there is no semantic independence constraint (i.e., $\tau = 0$), adding more dimensions to the optimum $r$ will not harm the representation power of $\boldsymbol{z}$. ii) If we only care about semantic independence and ignore the target task (i.e., $\tau \to 1$), the optimal $r$ would be equal to zero, indicating that a null representation is the best for discarding all semantic information. In this case, adding more dimension to $\boldsymbol{z}$ will necessarily violate the semantic independence constraint. More discussion can be found in the supplementary material.

In the following Theorem, we prove that the empirical solution converges to its population counterpart.

**Theorem 6.** Assume that $k_{\boldsymbol{s}}(\cdot, \cdot)$ and $k_{\boldsymbol{y}}(\cdot, \cdot)$ are bounded by one and $f_k^2(\boldsymbol{x}_i)$ is bounded by $M$ for any $k = 1, \ldots, r$ and $i = 1, \ldots, n$ for which $\boldsymbol{f} = (f_1, \ldots, f_r) \in \mathcal{A}_r$. For any $n > 1$ and $0 < \delta < 1$, with probability at least $1 - \delta$, we have

$$\Big| \sup_{\boldsymbol{f} \in \mathcal{A}_r} J(\boldsymbol{f}(\boldsymbol{x})) - \sup_{\boldsymbol{f} \in \mathcal{A}_r} J^{\mathrm{emp}}(\boldsymbol{f}(\boldsymbol{x})) \Big| \leq rM \sqrt{\frac{\log(6/\delta)}{a^2 n}} + \mathcal{O}\Big( \frac{1}{n} \Big),$$

where $0.22 \leq a \leq 1$ is a constant.

---

[5]The term 'solution' in any optimization problem in this paper refers to a global optima.

### 3.3 Trade-Off L

We recall that label space trade-off arises when the representation $z$ is ideal and is free to be designed optimally i.e., it does not necessarily depend on the input data $x$ or the encoder's hypothesis class. However, we assume that the representation $z$ is a direct effect of the target and sensitive variables ($y$ and $s$). Following [47], we use an additive noise model as

$$z = f_L(y, s) + e, \quad e \perp\!\!\!\perp y, e \perp\!\!\!\perp s \tag{15}$$

where $f_L(\cdot, \cdot) : \mathbb{R}^{d_y} \times \mathbb{R}^{d_s} \to \mathbb{R}^r$ is a Borel-measurable function. Following Section 3.1, we deploy $-\text{dep}(z, y)$, defined similar to $\text{dep}(z, s)$ in (8), as a proxy for the loss function $\inf_{g_Y \in \mathcal{H}_y} \mathbb{E}_{x,y}[\mathcal{L}_T(g_T(z), y)]$. Recall that, the desired optimization problem is given in (2). Instead of directly optimizing over $z \in L^2$, we optimize over all Borel-measurable functions $f_L(\cdot, \cdot)$ by ignoring $e$ since it is independent of both $y$ and $s$:

$$\sup_{f_L \in \mathcal{A}_r(y,s)} \Big\{ (1 - \tau) \, \text{dep}(f_L(y, s), y) - \tau \, \text{dep}(f_L(y, s), s) \Big\}, \tag{16}$$

where $\mathcal{A}_r(y, s)$ is defined similar to $\mathcal{A}_r$ in (10) by using $(y, s)$ instead of $x$ in the definition. Recall that $\mathcal{A}_r(y, s)$ ensures that $z$ will not contain highly correlated (entangled) dimensions, and thus be minimally redundant or maximally compact.

**Remark.** The optimization problem in (16) and its empirical counterpart can be solved similar to that of trade-off **D** in Theorems 5 and 6 where $x$ is replaced with $(y, s)$.

### 3.4 Trade-Off F

Here we define and discuss the trade-off achievable by practical realizations of representation learning algorithms with either fairness, invariance or semantic independence constraints.

**Definition 4.** *Feasible Space Trade-Off* arises from the statistical dependence between the target feature $y$ and the sensitive attribute $s$ conditioned on the given input data $x$, the choice of hypothesis class for the learners involved, and the choice of dependence measure adopted. This setting can be formalized as,

$$\inf_{f \in \mathcal{H}_x} \Big\{ (1 - \tau) \inf_{g_Y \in \mathcal{H}_y} \mathbb{E}_{x,y} \Big[ \mathcal{L}_Y \Big( g_Y(f(x)), y \Big) \Big] + \tau \, \widetilde{\text{dep}}(f(x), s) \Big\}, \quad 0 \le \tau < 1, \tag{17}$$

where $\mathcal{H}_x$ and $\mathcal{H}_y$ are the hypothesis class for the encoder network and target predictor, respectively, $\mathcal{L}_Y(\cdot, \cdot)$ denotes the loss function of target task, and $\widetilde{\text{dep}}(f(x), s)$ is a parametric or non-parametric surrogate measure of dependency quantifying the dependency between representation vector $z = f(x)$ and the sensitive attribute $s$.

This setting corresponds to the trade-off **F** in Figure 1(b), and is necessarily dominated by the Data Space Trade-Off **D**. Multiple factors may lead to such sub-optimal trade-offs. These include, hypothesis classes that are not universal RKHSs (e.g., [4] considered the case where $\mathcal{H}_x$ is universal, but $\mathcal{H}_s$ and $\mathcal{H}_y$ are linear RKHSs), the surrogate dependence measure $\widetilde{\text{dep}}(f(x), s)$ does not account for all non-linear dependencies (e.g., [3, 2, 21, 4] which consider adversarially learned dependence measures), sub-optimal optimization of (17) in terms of achieving only local optima but not the global optima (e.g., when the hypothesis class is deep neural networks that are optimized through stochastic gradient descent, or through stochastic gradient descent-ascent in the case of adversarial representation learning[3, 21, 2]), and combinations thereof.

## 4  Numerical Estimation of Trade-Offs

In this section, we demonstrate the practical utility of the analytical results developed in the paper and validate our theoretical insights. For this purpose, we design an illustrative toy example that conforms to the setting studied in the paper and numerically quantify the trade-offs that we introduced. Experimental validation on more tasks can be found in the supplementary material.

Consider the following Gaussian mixture model from which we generate $4000$, $2000$, and $2000$

$$v = [v_1, v_2] \sim \frac{1}{2} \Big( \mathcal{N}(m, \Sigma) + \mathcal{N}(m', \Sigma) \Big), \quad m = [0, 1], \; m' = [1, 1], \; \Sigma = \text{diag}(0.1^2, 0.1^2)$$

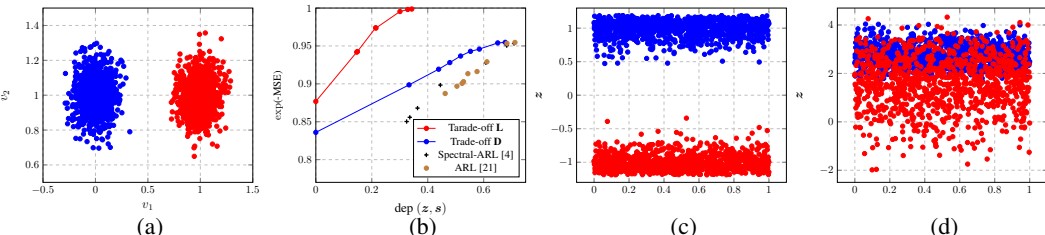

Figure 3: (a): A mixture of two Gaussians which generates the input data as $\boldsymbol{x} = v_1$, the sensitive attribute as $\boldsymbol{s} = v_1^3$, and the target attribute as $\boldsymbol{y} = [v_1, v_2^3]$. (b): Two fundamental trade-offs, **L** and **D**, together with two baseline feasible trade-offs **F**, ARL optimized with SGDA [21] and global optima of ARL with a linear RKHS [4]. (c), (d): The learned embedding for $\tau = 0$ and $\tau = 0.5$, respectively. An invariant representation should collapse $v_1$ i.e., the two colors should fully overlap with each other in the embedding. The overlap is partial for $\tau = 0.5$ and as $\tau \to 1$, the optimal representation is zero.

independent samples for training, validation and testing, respectively. Figure 3(a) shows the test samples where the samples generated with $\boldsymbol{m}$ and $\boldsymbol{m}'$ are in blue and red, respectively. The input data $\boldsymbol{x}$ is set to $v_1$ (the first entry of $\boldsymbol{v}$), the sensitive attribute $\boldsymbol{s}$ is $v_1^3$, and the target attribute $\boldsymbol{y}$ is $[v_1, v_2^3]$. In this problem both input data and target attribute are dependent on the sensitive attribute. We choose all three RKHS $\mathcal{H}_{\boldsymbol{x}}$, $\mathcal{H}_{\boldsymbol{y}}$, and $\mathcal{H}_{\boldsymbol{s}}$ to be Gaussian, which is a universal RKHS. The optimal $\boldsymbol{z}$ is learned for the trade-off **D** through the closed-form solution in Theorem 5 for different invariance parameter values $\tau$ in $[0, 1)$. Then, this optimal embedding is fed to a target task predictor which is a multi-layer perceptron (MLP) with two hidden layers, and $4, 8$ neurons and optimize the mean-squared error (MSE). The x-axis is a normalized version of the dependence measure used in our optimization, while the y-axis quantifies utility normalized to $[0, 1]$ as $\exp(-\text{MSE})$. The same procedure is implemented for trade-off **L**, except that the input data is $\boldsymbol{v}$, instead of $\boldsymbol{x}$. These trade-offs are shown in Figure 3(b). We choose the input data to be $\boldsymbol{v}$ instead of $(\boldsymbol{y}, \boldsymbol{s})$ for trade-off **L** since $(\boldsymbol{y}, \boldsymbol{s})$ is fully generated from $\boldsymbol{v}$ and therefore, $\boldsymbol{v}$ perfectly explains $(\boldsymbol{y}, \boldsymbol{s})$. For $\tau = 0$ and $\tau = 0.5$, the optimal embeddings are illustrated in Figure 3, (c) and (d), respectively. Since the sensitive attribute is only related to $v_1$, an invariant embedding should collapse the corresponding dimension and cause the two colors to overlap with each other.

We make the following observations, (a) Trade-off **L** dominates trade-off **D** as expected. (b) The trade-offs **F** obtained by the baselines are dominated by trade-off **D**. Adversarial representation learning [3, 21, 2] uses sub-optimal optimization (SGDA), while Spectral-ARL [4] uses a global optimum solution but restricts the hypothesis class in (3) to linear RKHS. As such, the baselines are unable to match the global optimal solution of (13), and (c) At $\tau = 0.5$ the embedding does indeed collapse $v_1$ to an extent leading to partial overlap between the two mixtures.

## 5   Conclusions and Societal Impact

This paper developed the theoretical underpinnings for identifying and determining the fundamental trade-offs and limits of representation learning under competing objectives. These trade-offs included i) label space trade-off which is solely induced by the statistical relation between target task and semantic attribute; ii) data space trade-off which is due to the statistical dependence between the input data and both target and semantic attributes. Further, we found closed-from solutions for the global optima, both the population and empirical versions, for the underlying optimization problems, and thus quantify the trade-offs *exactly*. Our results shed light on the regions of the trade-off that are feasible or impossible to achieve by learning algorithms. Numerical results suggest that commonly used adversarial representation learning based techniques are unable to reach the optimal trade-offs.

The theoretical results in this paper are useful for algorithmic fairness, privacy-preservation, and domain generalization applications of representation learning. Such systems are being widely deployed in a variety of practical applications: search engines, social media, law enforcement, healthcare, consumer devices, financial and judicial risk assessments, face analysis, and many more. Therefore, providing theoretical limits of performance is critically important for informed framing of regulatory policies, deployment of such solutions, and gaining societal trust. As such, we do not anticipate any adverse societal impacts from this work.

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
