# OpenReview forum: "On the Fundamental Trade-offs in Learning Invariant Representations"
_NeurIPS.cc/2021/Conference — NeurIPS 2021 Submitted_

### Official Review · Reviewer_q5TL · 2021-06-27

**Rating:** 5
**Confidence:** 4

**Summary:**

The paper proposes an MMD-style regularizer on the representation learned by a classifier in order to enforce invariance to a sensitive attribute. This is similar to existing work on invariant representation learning with an adversary, but using a measure of feature correlation which enforces more invariance than just the first-order statistics.

**Limitations And Societal Impact:**

Sure

**Main Review:**

I think this paper is decent, but not quite at the point where it's ready for publication.

There's quite a bit of overclaiming in the earlier writing when compared to the actual contribution presented. For example, the title of the work mentions the "tradeoffs and limitations" in invariant representation learning, and the earlier sections prepare the reader for some sort of new insight. However, upon a full read through, it is apparent that the tradeoffs and limitations to which the title refers is just the existing discussion on the "bottleneck principle" of Tishby et al, and the inherent tension between expressivity and invariance of a set of features when the label and sensitive attribute are correlated. This Pareto frontier is well-discussed in the literature (off the top of my head, see: "The cost of fairness in binary classification", "Costs and Benefits of Wasserstein Fair Regression" [which is concurrent with this submission, but further demonstrates that this perspective is not unique], in addition to all the works cited by this paper [8, 9, 10]).

Instead, the contribution of this paper (and what I believe the writing should more clearly focus on) is an MMD-like regularization penalty which under appropriate conditions correctly captures higher-order non-linear dependence between the learned features and the sensitive attribute in question. Theorem 1, while making an interesting observation about the limitations of adversarially invariant representation learning, is not the first time this has been pointed out (the authors acknowledge this). Furthermore, this is not an inherent fault of adversarial representation learning as claimed; rather, it is inherent to losses which are evaluated pointwise, such as MSE (the example used in the theorem statement). If, for example, we were to introduce a "critic" to regularize the distribution divergences (for example, to estimate mutual information or Wasserstein divergence), then adversarial representation learning would be able to identify higher-order dependencies. This is partially acknowledged, but much later (line 277), and it should be made clear earlier.

Ultimately, **I think the theoretical results for the proposed penalty term are valuable; I just think that there is too much focus on everything else, which is not newly presented in this work.** I should also note that I'm not very familiar with existing MMD-like divergence penalties for feature invariance. This general idea is being used in a lot of places: I've seen many such regularizers proposed, particularly recently, for OOD generalization [e.g., "Causally-motivated Shortcut Removal Using Auxiliary Labels", "An Information-theoretic Approach to Distribution Shifts", there are many more but I'm just picking a few to demonstrate that I'm not making this up]. As such, the overall novelty of this penalty term is questionable to me---it is possible that I'm just unaware of the previous literature.

Beyond these issues, the earlier notation of this work is very problematic:
* Are **x**, **y**, **z**, **s** fixed (after drawing the training distribution) or not? They are initially presented as random, but then the discussion of the label tradeoff treats the **x, y, s** as fixed (such that we can define the ideal **z** for each observation). And then **z** is presented as both fixed (chosen to be ideal for the particular training observation) *and* random (assumed to have a distribution with bounded second moment)?
* As far as I can tell, in the space of arbitrary measurable square-integrable functions f, shouldn't the label tradeoff be exactly equivalent to the data tradeoff? If **z** is assumed to have bounded second moment, then wouldn't the function f defined as f(**x**) = **z** satisfy square integrability? In which case, there always exists a function which achieves the "ideal" representation, regardless of the data distribution... I think the difference between the two in the experiments is actually the result of the "function tradeoff" idea, where such an f is undiscoverable.
* In the proof of Theorem 1, the distributional expectation of the loss is equated with the empirical loss over the training set. How does this make sense? The statement itself is trivial but it was surprisingly difficult to fully understand due to the notation.

Of course it's possible I'm misunderstanding all of these points, but I would argue that this is due to extremely inconsistent notation. I did not check the proofs in great detail because my functional analysis is not very strong, but these notational issues worry me.

I'm also unclear on why learning the "label-tradeoff" representation is at all meaningful? It requires access to the true label and sensitive feature, so of course it's useless in practice. I guess instead it's intended to show the gap between "label-tradeoff" and "data-tradeoff", but as I mentioned above I think this is actually an issue of estimation (finding the correct featurizer *f*) rather than one of approximation. In other words, I believe the "L -> D" gap as presented by the paper is actually a "D -> F" gap, and that the "L -> D" gap does not exist. If the authors disagree, it would go a long way to clean up the notation here.


--------------
**Update after rebuttal**

The authors have clarified some of my confusion, but I still feel that the emphasis of this work is too much on existing material and the material which is new is unclear and confusing, to the point that it requires significant revision. I am maintaining my score.

**Time Spent Reviewing:**

3.5

---

> ### Author Response · Authors · 2021-08-10
> **Existing Discussion of Trade-offs, MMD-like regularization, Notation confusion, Label vs Data trade-off**
>
> ## Title of Paper
>
>  We kindly remind that the title of our paper is "On the Fundamental Trade-offs in Learning Invariant Representations". It does not mention "limitations", though it mentions "trade-off".
>
> ## Existing Discussion of Trade-offs
>
>  We agree that the existence of trade-offs is well-known in the community (e.g., [8,9,10] and many more, including the bottleneck principal). However, our work seeks to go beyond acknowledging the existence of the trade-offs. Given the knowledge that the trade-off exists, we seek to answer, *Q1: What are the exact fundamental trade-offs that are possible between utility and invariance?* and *Q2: What is the encoder (mapping data to a representation) that achieves the fundamental trade-off and can we estimate it empirically?* To the best of our knowledge, (1) existing works only answers Q1 through upper and lower bounds on the trade-off, (2) existing works do not answer Q2.
>
> ## MMD-Like Regularization
>
> The aim of this paper is to primarily answer Q1 and Q2. Kernel measures of dependence provide us with the mathematical tools to do so, hence our choice of MMD-like regularization.
>
> ## Notation Confusion
>
> $\mathbf x, \mathbf y, \mathbf z, \mathbf s$​, all denote random vectors. The samples drawn from the distributions of these random vectors are denoted by $\mathbf x_i, \mathbf y_i, \mathbf z_i, \mathbf s_i$​, where "$i$​" is the sample index. Here, $\mathbf x_i, \mathbf y_i, \mathbf z_i, \mathbf s_i$​ are fixed. In label trade-off Section (Section 3.3), $\mathbf z$​ is still a random vector. To see why please look at equation (15), $\mathbf z=\mathbf f_L(\mathbf y, \mathbf s)+\mathbf e$​ where $\mathbf y, \mathbf s, \mathbf e$​ are all random vectors and $\mathbf f_L$​ is a Borel measurable function.
>
> ## Label vs Data Trade-Off Equivalency and Usefulness
>
>  There are two aspects to consider, existence and identifiability. The reviewer's comment acknowledges the latter but not the former.
>
> Let $\sigma(\mathbf x)$ be the $\sigma$-Algebra generated by the random vector $\mathbf x$. If we assume that $\mathbf z= \mathbf f(\mathbf x)$, then the $\sigma$-Algebra generated by $\mathbf z$ (i.e., $\sigma(\mathbf z)$) is always a subset of $\sigma(\mathbf x)$. In Trade-Off L, the unrestricted variable is $\mathbf{z}$ rather than $\mathbf{x}$. However, data $\mathbf{x}$ can be restricted, so there may not exist a valid encoder that can map $\sigma(\mathbf x)$ to $\sigma(\mathbf z)$. For example, let the ideal embedding $\mathbf z$ take values in $\{-1, 0, 1\}$ with identical probabilities (all $1/3$). Further, let the input data $\mathbf x$ be equal to $\mathbf z^2$. In this setting $\mathbf x$ take its values in $\{0, 1\}$ with probabilities of $1/3, 2/3$, respectively. In this case there is no measurable function $\mathbf f$ which can relate $\mathbf z$ as a function of $\mathbf x$ since $\sigma(\mathbf z)\not \subseteq \sigma(\mathbf x)$.
>
> To see the effect of this analysis intuitively, assume that $\mathbf x$​ does not provide sufficient information about the target attribute $\mathbf y$​ to perfectly predict it. In this case there will be neither a perfect predictor nor a perfect representation from $\mathbf x$​ to describe $\mathbf y$​. However, if we do not limit our representation $\mathbf z$​ to be a function of $\mathbf x$​, we can set $\mathbf z$​ to be equal to $\mathbf y$​. For the case of invariant representation learning, this idealization of $\mathbf z$​ is useful as an upper bound on the best possible trade-off achievable by *any* learning algorithm. We reaffirm that there will be a non-zero gap between label trade-off and data trade-off if data does not contain sufficient information to perfectly predict the target and semantic attributes.
>
> ## Typo in Proof of Theorem 1
>
> We mention that there are missing Expectation operator $\mathbb E_{\mathbf x, \mathbf s}$​ on the right hand side of equations below line 65 of the supplementary material. The final proof itself is unaffected by the typo. The summation is not over the training set, it is over each entry of the random vector $\mathbf s$​ (the $i$​-th entry/dimension of $\mathbf s$​ is denoted by $s_i$​).

---

> > ### Comment · Reviewer_q5TL · 2021-08-16
> > **Thanks for your clarifications**
> >
> > Apologies for misremembering the title.
> >
> > I think I understand now the "label tradeoff" vs "data tradeoff" idea. However, I think this submission still needs quite a bit of work in making this discussion more understandable, in addition to my earlier remarks on the fact that the focus of the writing appears to be less on the contributions of this work in particular, and more on analyses which are already known.
> >
> > To reiterate, I think the formal results shown here are interesting and potentially valuable, but the presentation needs quite a bit of refocusing and clarification. I would recommend relegating some of the formal discussion to the appendix, and instead working to carefully re-parameterize the formal statements and refocus the discussion to make these ideas more understandable. For example, the distinction between data vs label tradeoff makes sense in the way you've written it here (which took only a few sentences!), but I was unable to understand this concept after reading the entire original submission.

---

### Official Review · Reviewer_zJby · 2021-07-13

**Rating:** 5
**Confidence:** 2

**Summary:**

This paper identifies and determines two fundamental trade-offs between utility and semantic dependence induced by the statistical dependencies between the data and its corresponding target and semantic attributes. The authors derived closed form expressions for the global optima of the proposed optimization problems (for Tradeoff D, Tradeoff L and Tradeoff F among others), exposing the nature of the tradeoffs. Some convergence guarantees and experiments on synthetic data are also provided.

**Ethical Concerns:**

NIL

**Limitations And Societal Impact:**

This paper may have some impact in the study of fairness, privacy and other contemporary areas in machine learning.

**Main Review:**

Strengths:

+ Learning invariant representations and quantifying the tradeoffs between target and semantic attributes is an important topic in machine learning today in view of privacy protection and fairness.

+ The authors' formulation of these issues in terms of the mathematical optimization problems in Section 3 is rather neat. The fact that the solutions can be obtained in closed form is also very nice. It provides neat insights.

Weaknesses:

- The experimental section is rather weak. I would have expected a more extensive set of experiments, possibly on real-life data to be done. Given the extensive theory that has been developed, it is surprising that the authors have not exploited the theory more on real-world (or at least larger synthetic) datasets to corroborate for example the tightness of the bounds in Theorem 6. This is a missed opportunity. It would have been nice to experiment with various kernels. The dependence on the parameter \tau is also not well explained. It seems like a hyperparameter that is important in real-life applications.

Having said this, I must confess that I'm not an expert in this area of invariant representation learning.

**Time Spent Reviewing:**

2 hours

---

> ### Author Response · Authors · 2021-08-10
> **Experiment, Trade-Off Parameter**
>
> **Experiment:** For  an experiment on Adult dataset, please look at Section A.2 of the supplementary material where we compare our method with two state-of-the-art baseline methods. In this experiment we find the optimal representation and show the corresponding trade-off between accuracy of income classification and fairness w.r.t the race and gender..
>
> **Trade-off parameter:** The aim of this paper is to find utility and invariance for all trade-off parameter $\tau \in [0, 1)$. It is up to user to choose the appropriate $\tau$ from the full and exact trade-off frontier that we are providing.
> Please look at **General Comments** Section for more details.

---

### Official Review · Reviewer_aNqn · 2021-07-15

**Rating:** 3
**Confidence:** 3

**Summary:**

This paper studies the tradeoffs between accurately predicting the target and being invariant with respect to a known sensitive attribute in representation learning. When the features are functions in a RKHS, the paper derives closed-form solutions to the tradeoff optimization program. Finite sample guarantees are also given.

**Limitations And Societal Impact:**

Yes.

**Main Review:**

The paper tries to establish theoretical foundations for invariant representation learning, which is an important direction. The proposed method improves over baseline ARL. However, there are several major weaknesses:

1. The objective of optimization isn't well motivated. Why should one use HSIC as the dependence measure instead of mutual information?
2. The feature model is very restrictive, i.e. orthogonal kernel functions (eq 10) and the theory doesn't seem to be able to generalize to more commonly used models like neural networks.
3. The tradeoffs D, L, F seem to only differ in hypotheses classes. I don't see how the distinctions are meaningful.
4. No experiments on realistic datasets to demonstrate the usefulness of the theory.

**Time Spent Reviewing:**

3

---

> ### Author Response · Authors · 2021-08-10
> **Motivation of Optimization Problem, Orthogonal Kernel Functions, Trade-offs D, L, F, Numerical Experiment**
>
> ## 1.
>
> Calculating mutual information for high-dimensional continuous representation is computationally intractable and analytically challenging. Kernel methods for measuring of independence are an alternative solution with the attractive properties of being computationally feasible/efficient and analytically tractable. They are widely used in practice [4, 35, 37, 38, 39, 40, 45]. For more details please look at the **General Comments** Section
>
> ## 2.
>
> To give an intuitive view why orthogonal kernel functions (eq 10) are not restricting the representation, consider MSE loss for any representation $\mathbf z$​. This loss is a function of representation space (i.e., MSE loss is a function of orthonormal basis set spanned by the representation space). For simplicity consider a linear regression on top of the representation $\mathbf z$​. Let $\mathbf Z=[\mathbf z_1,\cdots, \mathbf z_n]$​ be all instances of the representation vector. Then, we have $\widehat{\mathbf y}=\mathbf W \mathbf z +\mathbf b$​, where the prediction loss is a function of projection on the subspace spanned by the rows of the matrix $\mathbf Z$​ (denoted by $P_{\mathbf z}$​) (equations 5, 6) of [4]. We can observe that orthonormalizing the matrix $\mathbf Z$​ does not effect $P_{\mathbf z}$​ and consequently, it does not change the loss of the linear regressor. Actually, disentanglement provides the representation vector with some desirable properties like non-redundancy and bounded variance. For more details of disentanglement please look at the **General Comments** Section.
>
> ## 3.
>
> The utility-invariance trade-offs depend on three factors: hypothesis class of encoder, dependence measure and optimization. Trade-off $\mathbf D$ is different from trade-off $\mathbf L$ if the data $\mathbf{x}$ is not  containing enough information to perfectly estimate the $\mathbf y$ and $\mathbf s$. On the other hand, trade-off $\mathbf L$ is independent of the imperfection of data and only depends on the joint distribution of $\mathbf{y}$ and $\mathbf{s}$. Trade-off $\mathbf F$ arises from either a restricted hypothesis class or sub-optimal optimization or incomplete dependence measure. This was stated in lines 274-281.
>
> ## 4.
>
> For experiment on realistic dataset, please look at the experiment on Adult dataset in Section A.2 of supplementary material.

---

> > ### Comment · Reviewer_aNqn · 2021-09-02
> > **Thank you for your response**
> >
> > Thank you for your response. I think the paper requires major revision so I will keep my score.

---

### Official Review · Reviewer_3Zq1 · 2021-07-16

**Rating:** 3
**Confidence:** 4

**Summary:**

This paper identifies and determines two tradeoffs between utility and dependence in learning invariant representations. The main contributions claimed by the authors are two folds: 1). closed-form solution for the global optima of the underlying optimization problem; 2). empirical estimates of the tradeoff terms that converge to the population counterpart in the large data limit.


**Limitations And Societal Impact:**

Yes.

**Main Review:**

Overall, I find this paper a little bit hard to follow. I think the overall presentation would be significantly improved if the authors could justify the claims made in the paper. For example, to motivate the problem, the authors claimed between Line 133-134 that "As such, ARL is inherently incapable of attaining the trade-offs achievable by complete measures of dependence". I personally think this is an overstatement, since the conclusion of Theorem 1 only applies to the mean-squared error in the regression setting, and conclusion of this theorem completely fails as long as one uses either the variational form of Wasserstein or TV distances as the score function of the discriminator. Hence I felt it is inappropriate to say that ARL is inherently incapable of attaining the trade-offs. In fact, it's just a matter on the choice of your adversarial loss function and the hypothesis class of your adversarial.

Perhaps my main concern is the lack of detailed comparisons with previous works and the overstatements of the contributions made in this work:
-   The authors mentioned in the related work that "In contrast to this body of work, our trade-off analysis is applicable to multi-dimensional discrete and/or continuous attributes where we find the exact optimal trade-offs." It seems to me that the conclusion in this paper only holds for continuous random vectors where the space of both S and Y are R. To see this, following the discussion between Line 164 to Line 170, the authors used Cov(\alpha(z), \beta(s)) where \alpha and \beta are in some RKHS. Per the original paper on HSIC [1], this criterion is for univariate continuous random variables, right?

[1]. Kernel Methods for Measuring Independence, https://www.jmlr.org/papers/volume6/gretton05a/gretton05a.pdf

-   A detailed comparison with a previous work is missing. In particular, this work makes the normality assumption of the joint distribution (Line 176 to 178) in the feature space and furthermore an independent assumption on the learned features (Eq. 10) in order to derive the tradeoff result. Similarly, [2] also makes the normal assumption but is free of the disentanglement assumption. In terms of technical results, it appears that the results in [2] (in the regression section) give similar bounds, i.e., the optimal representations are given by the top two eigenvectors of the kernel covariance matrix, and the authors also show that under the Gaussian assumption their lower bound becomes tight. I was wondering what's the difference between the results of this paper to that of [2]? Given the close relationship between these two works, the authors should make a clear cut comparison w.r.t. the bounds in [2]. So far such a comparison is completely missing.

[2]. Fundamental Limits and Tradeoffs in Invariant Representation Learning, https://arxiv.org/abs/2012.10713

-   The authors claimed that their closed-form empirical estimators lend themselves to practical invariant representation learning algorithms. Where are the proposed invariant representation learning algorithms in the paper? It seems to me that the experiments are mainly used to verify the theoretical results in the paper. This is totally fine, but the claims made in the introduction does not reflect what's going on in the experiment section.

Furthermore, there are several claims in the paper that are not justified and do not seem correct to me, including:
-   Line 33, the authors claim that "This independence condition is satisfied if and only if (iff), the representation z is independent of s". I don't think this is true. To see a simple counterexample, it's certainly possible that z is not independent of s while the predictor \hat{y} being independent of s -- simply making \hat{y} to be a constant function of s.

-   Line 50-51, the authors claim that "Including all measurable functions in the hypothesis class of the encoder f(·) and target predic51 tor gY (·) ensures that the best possible trade-off is included within the feasible solution space." I was wondering why this is true? Presumably, even if one works with all the Borel-measurable functions, these functions are still determnistic mappings. Would this be sufficient to achieve all the feasible solutions?

-   Line 192-193: could the authors further elaborate on why this statement is true? "Using, the representer theorem [45], it follows that f(x) = ΘE[kx(x1, x), · · · , kx(xn, x)]T, where Θ ∈ R r×n is a free parameter matrix."

===============================================================

I have read the authors' responses as well as other reviews. Please see my detailed comments below. Overall, I am not convinced by the response and I will adjust my rating from 4 to 3.


**Time Spent Reviewing:**

4

---

> ### Author Response · Authors · 2021-08-10
> **Comparison to [10], Disentanglement, Practical Learning Algorithm, Justifications of Some Claims**
>
> ## Multidimensionality of Our Setting
>
> In $\text{Cov}(\alpha(\mathbf z), \beta_{\mathbf s}(\mathbf s))$​ and $\text{Cov}(\alpha(\mathbf z), \beta_{\mathbf y}(\mathbf y))$​ the domain of $\mathbf y$​ and $\mathbf s$​ can be any compact set including a discrete set or a continuous subset of $\mathbb R^d$​ for arbitrary $d$​ [37]. For example, we recall RBG-Gaussian kernel where we have
>
> $$
> \beta_{\mathbf y} (\mathbf y) = \sum_{i=1}^n w_i\exp(-\frac{d(\mathbf{y}_i,\mathbf{y})}{2\sigma^2}).
> $$
>
> We can observer that $\mathbf y$​ can be discrete or multidimentional continuous. Same reasoning holds for $\alpha(\mathbf z)$​ and $\beta_{\mathbf s}(\mathbf s)$​.
>
> ## Comparison to [10]
>
> Firstly, please look at the response to reviewer **9fPS** for primary comparison of this work with [10]. We recall that the optimal representation for upper bound ( in [10] is given by top two eigenvectors of  the operator
>
> $$
> R=\Sigma_{xx}^{1/2} (\lambda aa^T -y y^T)\Sigma_{xx}^{1/2}
> $$
>
> where $a$ and $y$ are optimal Bayes estimators for semantic and target attributes, respectively. Obviously the rank of $R$ cannot be more than $2$ due to the existence of $(\lambda aa^T -y y^T)$ in $R$. This also explains why $\Sigma_{xx}^{1/2} ( aa^T)\Sigma_{xx}^{1/2}$ cannot capture all statistical dependency between the representation and the semantic attribute:  $( aa^T)$​ restricts the covariance operator to only look for mean independency (i.e., the dependency imposed by the direction of Bayes estimator). In contrast, the optimal representation in our paper (Theorem 5 and Corollary 5.1) is all  eigenvectors corresponding to the non-negative eigenvectors of
>
> $$
> \Big((1-\tau)\Sigma^*_{\mathbf y \mathbf x}\Sigma_{\mathbf y \mathbf x} -\tau\,\Sigma^*_{\mathbf s \mathbf x}\Sigma_{\mathbf s \mathbf x}\Big) f = \lambda \, (\Sigma_{\mathbf x \mathbf x} +\gamma I) f. \nonumber
> $$
>
> Using the notation in [10], solution to a simplified formulation of our trade-off reduces to
> $$
> B:=\Sigma_{xx}^{1/2} \Big(\lambda \sum_{i} \theta_i\phi_i(A)\phi_i(A)^T - \sum_{j} \theta_j\psi_j(Y)\psi_j(Y)^T\Big)\Sigma_{xx} ^{1/2}​,
> $$
> where $\phi_i$​​ and $\psi_j$ are non-linear mappings induced by some RKHSs. Clearly rank of $B$ is not two. Specifically, rank of $B$​​ is not degenerated by the direction(s) imposed by any loss function and can capture all directions of dependency.
>
> **Disentanglement:** The Disentanglement condition is actually implicitly considered in [10]. The MSE loss for any representation $\mathbf z$​ is a function of representation space (i.e., MSE loss is a function of orthonormal basis set spanned by the representation space). This can be observed in the proof of Theorem 5.1 in [10] (in the top of page 28 of [10, version1]). For simplicity consider a linear regression on top of the representation $\mathbf z$​. Let $\mathbf Z=[\mathbf z_1,\cdots, \mathbf z_n]$​ be all instances of the representation vector. Then, we have $\widehat{\mathbf y}=\mathbf W \mathbf z +\mathbf b$​, where the prediction loss is a function of projection on the subspace spanned by the rows of the matrix $\mathbf Z$​ (denoted by $P_{\mathbf z}$​) (equations 5, 6) of [4]. We can observe that orthonormalizing the matrix $\mathbf Z$​ does not effect $P_{\mathbf z}$​ and consequently, it does not change the loss of the linear regressor. Actually, disentanglement provides the representation vector with some desirable properties like non-redundancy and bounded variance. For more details of disentanglement please look at the **General Comments** Section.
>
> ## Practical Learning Algorithm
>
> For practical invariant representation learning algorithm, we firstly refer to Theorem 5 of our paper where the optimal encoder is the eigenvectors of a generalized eigenvalue problem in equation (14):
> $$
>         \Big(\mathbf L^T_{\mathbf x}\big((1-\tau)  \mathbf K_y  -\tau \mathbf K_s\big)\mathbf L_{\mathbf x}\Big) \mathbf u = \lambda \Big(\mathbf L^T_{\mathbf x} \mathbf H \mathbf L_{\mathbf x} + n\gamma \mathbf I\Big) \mathbf u\nonumber.
> $$
> Note that all involved kernel matrices can be calculated from the training set. The Pytorch implementation of this empirical calculation can be found in the supplementary material. In addition to a toy example in Section 4, we also have experiments on Adult dataset in Section A.2 of supplementary material where we compare our method with two state-of-the-art baseline methods.
>
> ## Justification of Claims
>
> **For the justification of the claims in line 33:** "This independence condition is satisfied if and only if (iff), the representation $\mathbf z$​​​ is independent of $\mathbf s$​​​", we refer the reviewer the sentences preceding it i.e., lines 31-33: "Invariant learning requires that prediction of the target label, $\widehat{\mathbf y} = g_Y(\mathbf z)$​​​ be independent of the semantic attribute $\mathbf s$​​​  for all possible downstream target predictors $g_Y(\cdot)$​​​. This independence condition is satisfied if and only if (iff), the representation $\mathbf z$​​​ is independent of $\mathbf s$​​​". The key is *all possible downstream targets*. The example given by the reviewer ($\hat{y}=c$​​​) may work for a specific instance of downstream task $\mathbf y$​​​ but not all downstream tasks.
>
> **Borel measurable functions:** Let $\mathbf x$​​​​​​​​​​​​​ be a random vector ($d$​​-dimensional) in the probability space $(\Omega, \mathcal F, P_x)$​​ where $\Omega$​​ is the sample set, $\mathcal F$​​ a $\sigma$​​-algebra defined on $\Omega$​​ and $P$​​ a probability measure on $\mathcal F$​​. Consider any function $f(\cdot):\mathbb R^d\rightarrow \mathbb R^r$​​. In order for $\mathbf z=f(\mathbf x)$​​ to be a valid random vector on the probability space $(\mathbb R^{d}, \mathcal B, P_z)$​​, $f(\cdot)$​​ should to be a Borel measurable function. Even if $f_u(\cdot)$​​ is a randomized function w.r.t. the random variable $u$​​, for each instance $u_0$​​ of $u$​​, $f_{u_0}(\cdot)$​​​​​​​​​​​​​​ should be a Borel measurable function.
>
> **Representer Theorem:** Representer theorem states that any function $f(\mathbf x)$ in the RKHS $\mathcal H_{\mathbf x}$ can be expressed as a linear combination of the basis functions $\{k_{\mathbf x}(\mathbf x_1, \mathbf x), \cdots, k_{\mathbf x}(\mathbf x_n, \mathbf x)\}$, i.e., $f(\mathbf x)=\sum_{i=1}^n \theta_i k_{\mathbf x}(\mathbf x_i, \mathbf x)$, where $[\theta_1,\cdots,\theta_n]^T$ is a free vector in $\mathbb R^n$. Now, since our $\mathbf f=[f_1, \cdots, f_r]$ where $f_1,\cdots,f_r$ all belong to $\mathcal H_{\mathbf x}$, therefore, $\mathbf f(\mathbf x)=\mathbf \Theta [k_{\mathbf x}(\mathbf x_1, \mathbf x),\cdots,k_{\mathbf x}(\mathbf x_n, \mathbf x)]^T$ where $\mathbf \Theta\in \mathbb R^{r\times n}$ is a free matrix.

---

> > ### Comment · Reviewer_3Zq1 · 2021-08-29
> > **Follow-up**
> >
> > First, I'd like to thank the authors for providing a detailed response. However, I must have to admit that I am confused by the response: "...due to the usage of MSE and cross-entropy losses for adversary...". I don't think this is true. Cross-entropy is an upper bound of the conditional entropy, so when used as the loss function with a rich set of adversaries, e.g., all the measurable functions, this naturally leads to the use of mutual information as a characterization of the dependency measure, which clearly is equivalent to independence between random variables.
> >
> > Unfortunately, the argument that "This also explains why ... cannot capture all statistical dependency between the representation and the semantic attribute:" is also not true. It is true that the rank of the matrix is at most 2, but it has nothing to do with the dependency per se. Rather, rank 2 simply follows from the fact that both the target and the sensitive attributes are scalar outputs. In [10], if we consider multi-dimensional target and sensitive attributes as in the setting of this paper, then the rank would be more than 2.
> >
> > In the authors' response to Reviewer 9fPS, the authors mentioned that "The universality condition on H is necessary, otherwise H may not contain (at least in the limit) the Bayes estimators...". This is not true. Please take a look at Assumption 5.1 in [10]. The existence is assumed by the assumption, hence the above argument is irrelevant. Of course, assuming H to be universal is also fine, since it's a stronger assumption that is sufficient to guarantee existence.
> >
> > Overall, I have to admit that I am not convinced by the authors' response. It seems to me that the authors also have a significant misunderstanding about prior works. Hence, I will adjust my rating from 4 to 3.

---

> > > ### Author Response · Authors · 2021-09-01
> > > **More Clarifications**
> > >
> > > **MSE Loss:**
> > > Please kindly look at Theorem 1 (line 127 of our paper) and references [32], [33] for proof that MSE loss for adversary only accounts for mean dependency not all kinds of dependency. To give an intuitive example consider the case where the sensitive attribute $\mathbf s$ is a uniform random variable on $[-1, 1]$ and the representation is $\mathbf z$ is equal to $\mathbf s^2$. Obviously, $\mathbf z$ is totally dependent on $\mathbf s$. Consequently, if $\mathbf z$ is given to be $\mathbf z_0\in [0, 1]$, then it follows that
> > > $$
> > > \mathbb E[\mathbf s | \mathbf z =\mathbf z_0]= 0.5 \sqrt{\mathbf z_0} - 0.5 \sqrt{\mathbf z_0}=0.
> > > $$
> > > Therefore, the optimal Bayes regressor will results to $\mathbb E[\mathbf s|\mathbf z]=0$ for all values of $\mathbf z\in [0, 1]$. On the other hand, if $\mathbf z$ is independent of $\mathbf s$, the optimal Bayes regressor is $\mathbf E[\mathbf s]$ which is zero again. Therefore, the optimal Bayes regressors in both cases (where $\mathbf z=\mathbf s^2$ or $\mathbf z$ is an independent random variable of $\mathbf s$), is predicting zero. The implication is that maximizing MSE loss over $\mathbf z$ does not necessarily learn a representation that is statistically independent of $\mathbf s$, rather learn a representation that is mean independent of $\mathbf s$. As a result, the formulation in [10] (regression setting) does not account for the fundamental trade-off in invariant representation learning.
> > >
> > > **Multidimensional Target and Sensitive Attributes**:
> > > We respectfully disagree with the reviewer that the rank of the matrix
> > > $$
> > > R=\Sigma_{xx}^{1/2} (\lambda aa^T -y y^T)\Sigma_{xx}^{1/2}
> > > $$
> > > in [10] has no thing to do with the dependency between $\mathbf z$ and $\mathbf s$.
> > > Please note that if the dimension of $\mathbf s$ is $d$,
> > > then the rank of $R$ in [10] would be at most $2d$ . However, still, $R=\Sigma_{xx}^{1/2} ( aa^T)\Sigma_{xx}^{1/2}$ is degenerating the rank of $\Sigma_{xx}$ and therefore
> > > $R=\Sigma_{xx}^{1/2} ( aa^T)\Sigma_{xx}^{1/2}$ is not able to characterize all modes of dependency between representation $\mathbf z$ and the sensitive attribute $\mathbf s$. Unlike [10], please observe that in our setting (Theorem 4), statistical dependency is characterized by
> > > $$
> > > \Sigma^*_{\mathbf s \mathbf x}\Sigma_{\mathbf s \mathbf x},
> > > $$
> > > where its rank is infinite and it captures all modes of dependency between $\mathbf z$ and $\mathbf s$.
> > >
> > > **Cross-Entropy Loss:**
> > > We acknowledge that for **binary** $\mathbf s$, using cross-entropy loss leads to the use of mutual information which fully characterizes the statistical dependency between $\mathbf s$ and $\mathbf z$.  However, for multidimensional $\mathbf s$, cross-entropy no longer captures all modes of dependency (see [32], below Eq. 1).
> > > If the reviewer disagrees about this statement, we kindly request her/him to provide some proofs or references that cross-entropy loss for multi-class $\mathbf s$ can capture all modes of statistical dependency between $\mathbf z$ and $\mathbf s$.
> > >
> > > We argue that mutual information should be directly used as Eq. (9) in [10] instead of cross-entropy loss for multi-dimensional $\mathbf s$. However, mutual information is computationally intractable for high dimensional $\mathbf z$ when $\mathbf s$ is also multidimensional. In contrast to [10], we are using kernel method which is theoretically applicable to all cases where $\mathbf s$ and/or $\mathbf y$ are binary, multi-class, or multidimensional continuous and computationally tractable for high dimensional $\mathbf z$.
> > >
> > > **Assumption 5.1 in [10]:**
> > > We acknowledge that Assumption 5.1 was overlooked by us. However, as we clearly mentioned in our response to reviewer **9fPS**, our comparison to [10] is assuming that all technical details in [10] are valid. Therefore, missing Assumption 5.1 in [10] by us does not affect our comparison with [10] at all.

---

### Official Review · Reviewer_9fPS · 2021-07-22

**Rating:** 4
**Confidence:** 4

**Summary:**

This paper studies the utility-invariance trade-off problem and characterizes closed-form expressions for two types of trade-offs. This is an important problem since it's relevant to privacy, fairness and representation learning.

**Ethical Concerns:**

-

**Limitations And Societal Impact:**

-

**Main Review:**

This paper studies the utility-invariance trade-off problem and characterizes closed-form expressions for two types of trade-offs. This is an important problem since it's relevant to privacy, fairness and representation learning. The results are theoretically sound and the paper is mostly well-written.

However, there are a few aspects that the current version of the work is lacking, detailed below.

Comparison to [10]: The authors did not provide much comparison with the most closely related work [10]  (the only relevant part I found is line 96-98).  Both of these two papers work on very similar problem with only a few small differences in the choice of loss functions, and the results are similar from high level as well (and they even have nearly identical titles). While some comparison will be mentioned in this review, I think it's necessary to include a thorough comparison in the future version of this work.

Clarity: This paper is mostly well-written. However, there are some mathematical details that are not clearly explained.

(1) In equation (10), the definition of the constrained function space $A_r$ involves an additional (tunable) parameter $\gamma$, whose role is not discussed in detail (apart from mentioning it as a trade-off parameter), nor is there any theoretical/empirical suggestion for how to choose the parameter. Furthermore, the main result (Theorem 4) holds regardless how $\gamma$ is chosen - which appears to be strange since $A_r$ itself depends on $\gamma$.

(2) In Definition 1 & Definition 2, the trade-off objectives are stated as a linear combination of (a) prediction loss on y and (b) dependence between z and s. However, in the main theorem (Theorem 4), the part (a) is replaced with dependence between z and y. While the authors mentioned in line 213 and line 129-134 some explanations about this change, this not only causes confusion for the reader, but also (to some extent) hides a flaw in the reasoning about the design choice of loss, detailed below.

Choice of the loss function and Technical difficulty: As mentioned above, the authors replaced the target loss with the dependence (HSIC) loss. But this choice is poorly justified - while it was mentioned in Theorem 1 that the MSE loss does not guarantee statistical independence, this reasoning only applies to the sensitive feature ($s$) part, where one hopes to achieve independence with The same reasoning does apply to the target loss $y$ part, where the goal is to make accurate prediction, rather than strong dependence.

Furthermore, the improvement over [10] mentioned in line 99-102, are mostly due to this change - the original formulation in eqn (1) appears to be much more difficult to deal with comparing to the setting in theorem 4, while theorem 4 bypasses this difficulty and only require simple eigenvector/eigenvalue analysis. The weaker characterization in [10] (only lower bounds are provided there) involves significantly more challenging analysis, for which I suspect that the authors would encounter similar challenge if they didn't made this simplification.

Experiments: The experiment section is very short and only toy dataset  (mixture of two Gaussians) is included. It is also unclear how the theory would impact the applications mentioned in the paper like domain adaptation and fairness.



**Time Spent Reviewing:**

8

---

> ### Author Response · Authors · 2021-08-10
> **Comparison to [10], HSIC for the Target Task**
>
> ## Typo in Theorem 4
>
> There is a typo in the statement of Theorem 4. The right side of equation (12) should be $\lambda (\Sigma_{\mathbf x\mathbf x} + \gamma I) f$​​. However, in the proof of Theorem 4, this typo does not exist (please look at line 124 of supplementary material).
>
> ## Comparison to [10]
>
> Below we compare with version1 of [10], since version 2 (latest) appeared on 23 July 2021 which is after NeurIPS-2021 paper submission deadline.
>
> Firstly, some proofs (specifically Theorem 5.3) in [10] are not mathematically rigorous and it is difficult to check their correctness. For example, Assumption 5.1 refers to "the RKHS $\mathbb H$" but $\mathbb H$ is not clearly defined. However, we will assume that $\mathbb H$ is an arbitrary universal RKHS (e.g., induced by RBF-Gaussian). The universality condition on $\mathbb H$ is necessary, otherwise $\mathbb H$ may not contain (at least in the limit) the Bayes estimators, ${f^*}_Y$ and ${f^*}_Y$.
>
> Similarly, at the bottom of page 11 the covariance operator is defined as $\Sigma:=\text{Cov} (\phi (\mathbf x), \phi(\mathbf x))$ without defining the corresponding domain and co-domain of this operator, as well as the action of this operator. We guess that $\Sigma$ is a bi-linear functional from $\mathbb H \times \mathbb H$ to $\mathbb R$ and $\langle f_1, \Sigma f_2\rangle_{\mathbb H}=\text{Cov}(f_1(\mathbf x), f_2(\mathbf x))$ for any $f_1, f_2\in \mathbb H$.
>
> Assuming that all theoretical findings in [10] are correct, below are the main limitations of [10] which do not exist in our work:
>
> * **Dependence Measure:** Statistical dependence between the representation $\mathbf z$ and the semantic attribute $\mathbf s$ in [10] is modeled via an adversary optimizing an MSE or cross-entropy losses. As it has been shown in Theorem 1 of our paper and [32, Section 3], such dependence measures cannot capture all kinds of statistical dependencies. So this trade-off formulation in [10] is similar to Trade-Off F in our submission. In contrast, our formulation is capable of capturing  all kinds of dependencies through a universal kernel measure of dependency.
>
> * **Characterizing Trade-Off:** [10] finds an upper bound (might be tight but still a bound and not exact). The bound does not involve data $X$​ and is defined in terms of $Y$​ and $S$​. As such, the bound is similar to Trade-Off L in our submission. In contrast we are able to characterize two utility-invariance trade-offs exactly in a closed-form expression. Our Trade-Off D is obtained by the optimal encoder under hypothesis class of all measurable functions where the input data may not contain sufficient information to perfectly predict the target and semantic attribute, and Trade-Off L corresponds to _any_ unrestricted learner who is provided with perfect information about target and semantic attributes.
>
> * **Optimal Encoder:** [10] provides a bound on the trade-off, but does not find the encoder that realizes neither the exact trade-off nor its bound. In other words, there is no algorithm to obtain optimal (or near optimal) representation learning neither in population nor in empirical settings. Therefore, the theoretical analysis in [10] do not directly lend themselves to any efficient algorithm for representation learning. In contrast, we proposed an empirical procedure  (Theorem 5 and Corollary 5.1) to numerically obtain the optimal encoder in terms of the eigenvectors of a generalized eigenvalue problem in equation (14):
>   $$
>  \Big(\mathbf L^T_{\mathbf x}\big((1-\tau)\mathbf K_y  -\tau \mathbf K_s\big) \mathbf L_{\mathbf x}\Big)\mathbf u
> = \lambda \Big(\mathbf L^T_{\mathbf x} \mathbf H \mathbf L_{\mathbf x} + n\gamma \mathbf I\Big) \mathbf u
>   $$
>
> * **Random Variables:** Both target attribute $\mathbf y$ and semantic attribute $\mathbf s$ are restricted to be continuous or discrete at the same time (e.g., it is not possible to have $\mathbf y$ continuous while $\mathbf s$ is discrete). In contrast, in our work $\mathbf y$ and $\mathbf s$ can be of different nature, discrete and/or multidimensional continuous random variables.
>
> * **Target Loss:** [10] Considers the MSE as a target loss in regression setting, while we instead optimize HSIC. As we describe in **General Comments** (Theorem there) , HSIC for representation learning is equivalent to MSE if a linear RKHS for the target attribute is deployed . More generally, as we argued in the **General Comments**, optimizing HSIC can ensure that the representation $\mathbf{z}$​​ contains sufficient information that can be exploited by a downstream regression or classification tasks.

---

### Author Response · Authors · 2021-08-10
**General Comments**

## Contributions

We would like to summarize and reiterate the contributions of our submission.

We seek to answer the following questions:

*Q1: What are the exact fundamental trade-offs that are possible between utility of the representation and its invariance?*

*Q2: What is the encoder (mapping data to a representation) that achieves the fundamental trade-off?*

Existing theoretical analysis of the utility-invariance trade-off seeks to answer Q1 e.g., under surrogate measures of invariance such as adversarial learning optimizing MSE or cross-entropy [10] to estimate bounds on the trade-off under a general function class. And, there is no existing work that answers Q2 under a general setting in terms of complete measure of invariance and a general function class.

Our submission answers both Q1 and Q2. We formulate the problem of learning an encoder as an optimization problem with a complete measures of dependence (a slight variant of HSIC) and a general hypothesis space (Borel measurable functions) for the encoder. Then, we provide a closed-form solution for the global optima of the optimization problem i.e., find the optimal encoder corresponding to Q2. Finally, we use this solution to find the exact fundamental trade-off i.e., Q1. Additionally, we describe an empirical estimator for the optimal encoder and corresponding trade-off.

## Inherit incapability of ARL as A Dependency Measure

We acknowledge that the conclusion (lines 132-134) of Theorem 1 is not accurate and agree with the reviewers that ARL with an appropriate loss function can be a complete measure of dependence. We will modify the statement to say that ARL optimizing MSE loss is not a complete measure of dependence. Furthermore, it has been shown that [32, Section 3] cross-entropy loss is also incapable of capturing all kinds of dependency.

We do note that the theoretical analysis in [10] corresponds to ARL where in both regression and classification settings it does not characterize the fundamental utility-invariance trade-off due to the usage of MSE and cross-entropy losses for adversary.

## More Elaboration on $\mathcal{A_r}$

Recall the definition of
$$
\mathcal A_r= \Big\\{ \big(f_1,\cdots, f_r \big)\ \Big| \ f_i, f_j\in \mathcal H_{\mathbf x},\  \text{Cov} \big(f_i(\mathbf x), f_j(\mathbf x) \big) + \gamma \langle f_i, f_j\rangle =\delta_{i,j} \Big\\} \nonumber.
$$
$\mathcal A_r$ consists of two parts:

1. The covariance between different dimensions of the embedding $ \mathbf z=[f_1(\mathbf x), \cdots, f_r(\mathbf x)]^T$. This part encourages the covariance matrix of the embedding vector to be identity matrix. This kind of disentanglement has been used in independent component analysis (ICA) [39]. It enables us to ensure that the variance of each entry of $\mathbf z$ is bounded and different entries of $\mathbf z$ are uncorrelated to each other.
2. The $\langle f_i, f_j\rangle $ part encourages that the functions of representation be as orthogonal as possible to each other and to be of unit norm. This part prevents the representation from being redundant (e.g., dimensions with repeated entries) and helps with numerical stability during empirical estimation. Note that this concept is identical to the orthonormality constraint on the projection matrix in PCA.

Each of above parts correspond to different notions of disentanglement and by linearly combining them, we are designing a flexible mechanism which can be tuned by the user, depending on the application at hand. In Section A.1 of supplementary (line 17) we suggest that $\gamma$ can be tuned through cross-validation.

## Justification of HSIC As a Surrogate for Target Loss

**We firstly point out that the primary problem in our paper is representation learning. Maximizing the statistical dependency between representation vector and the target attribute can flexibly learn a representation that can be effective for different downstream target tasks, including, regression, classification, and etc. We support our arguments in the following.**

Consider the case where the invariance to the semantic attribute $\mathbf s$ is ignored and only the representation learning for the target attribute $\mathbf y$ is considered.  This setting is corresponding to the standard supervised representation learning and based on Theorem 4, it is equivalent to finding eigenvectors corresponding to $r$ largest eigenvalues of the following generalized problem
$$
\big(\Sigma^*_{\mathbf y \mathbf x}\Sigma_{\mathbf y \mathbf x}\big) f = \lambda  \big(\Sigma_{\mathbf x \mathbf x} +\gamma I\big) f. \nonumber
$$
This optimization problem is a variant of kernelized supervised PCA where it has been shown to be effective on both regression and classification problems [A].  Recently, HSIC  has also been successfully used in self-supervised representation learning where [B] demonstrated that maximizing HSIC between representation vector and the image identity results in effective representation that is useful for different downstream vision tasks.

Specifically, for the regression task we present the following Theorem.

**Theorem:** *Let $\mathbf x \in\mathbb {R}^{d_x}$​​​​​ and $\mathbf y\in\mathbb {R}^{d_y}$​​​​​ be multivariate random vectors corresponding to the input data and the target attribute, respectively. Let  $\mathcal H_{\mathbf x}$​​​​​ be an arbitrary universal RKHS and $\mathbf f =[f_1,\cdots, f_r]^T$​​​​​ where $f_i$​​​​​'s all belong to $\mathcal H_{\mathbf x}$​​​​​.​ Let $\mathcal H_{\mathbf y}$​​ be a linear RKHS and assume that $\big(\mathbf f(\mathbf x),\mathbf y\big)$​​​​​​​​​​ is a jointly Gaussian random vector. It has been shown in [4] (Section 3.3) that minimizing the following representation learning for regression problem*
$$
\min_{\mathbf f\in \mathcal H_{\mathbf x}}\ \min_{g_Y(\cdot)\, \text{measurable}} \mathbb E_{\mathbf{x, y}} \Big[\big\\| g_Y(\mathbf f(\mathbf x))-\mathbf y\big\\|^2\Big]
$$

*is equivalent to finding eigenvectors corresponding to $r$​​​​​ largest eigenvalues of the following problem*
$$
\big(\Sigma^*_{\mathbf y \mathbf x}\Sigma_{\mathbf y \mathbf x}\big) f = \lambda  f \nonumber
$$
*Furthermore, this is equivalent to our formulation in (12) with a slight change in $\mathcal A_r$  as*
$$
\mathcal A_r=\Big\\{\big(f_1,\cdots, f_r \big)\ \Big| \ f_i, f_j\in \mathcal H_{\mathbf x},\
    \langle f_i, f_j\rangle =\delta_{i,j}\Big\\} \nonumber.
$$
*In this variant of $\mathcal A_r$​​​, only orthonormality is considered and uncorrelatedness is ignored.*

The above Theorem states that minimizing the MSE loss for representation learning is exactly equivalent to maximizing the dependency of the representation to the target attribute when  $\mathcal H_{\mathbf y}$​​ is a linear RKHS.

[A] E. Barshan, A. Ghodsi, Z. Azimifar, and M. Zolghadri Jahromi, "Supervised principal component analysis: Visualization, classification and regression on subspaces and submanifolds", Pattern Recognition, 2011.

[B] Y. Li, R. Pogodin, D. Sutherland, A. Gretton, "Self-Supervised Learning with Kernel Dependence Maximization", arXiv preprint arXiv:2106.08320, 2021.

## Numerical Experiment

Experiments on the Adult dataset were provided in Section A.2 of supplementary material where we compare our method with two state-of-the-art baseline methods. In this experiment we find the optimal representation and show the corresponding trade-off between the accuracy of income classification and fairness w.r.t the race and gender.

---

### Decision · Program_Chairs · 2021-09-27

**Decision:**

Reject

**Comment:**

This paper analyzes various tradeoffs in invariant representation learning. During review, several concerns have been raised including missing comparisons with closely related work, clarity and exposition, and some misunderstandings at a conceptual level. These concerns were not adequately addressed during the discussion phase. In the end, there was a consensus amongst the reviewers that this paper is not ready for publication in its current form.